# Quantifying Hallucination Bias in AI Generated Deepfakes: A Multimodal Analysis Using Divergence Metrics

Mohak Dwarkadhish Sharma ⬤
Anshita Bhardwaj
School of Computing and Augmented Intelligence
Arizona State University
Tempe, United States of America
e mail: mshar118@asu.edu, anshita.inbox@gmail.com

*Abstract*—Hallucinations and deepfakes represent two emergent challenges in generative AI. Hallucinations arise from model overfitting or misinterpretation, causing AI systems to produce content with no basis in reality, whereas deepfakes are deliberately generated synthetic media crafted to mimic real world data. This paper hypothesizes that an overfitted generative model (hallucination prone) can exhibit output deviations analogous to or exceeding those of a well regularized deepfake model. To investigate, we train two convolutional autoencoders on the FaceForensics++ dataset one overfit on authentic data (to induce hallucinations) and one regularized on manipulated data (to emulate deepfakes) and introduce a novel divergence metric $\theta$ to quantify their output differences. $\theta$ is defined as the ratio of reconstruction errors between the hallucination model and the deepfake model for the same input. A comprehensive evaluation is conducted, including classification performance, distribution of $\theta$, latent space visualizations, and statistical tests. We further enhance the analysis with well known divergence metrics (Kullback–Leibler divergence, Fréchet Inception Distance, Structural Similarity Index) to contextualize $\theta$. Experimental results show that both models achieve high face classification accuracy, yet the hallucination prone model's reconstructions diverge significantly more from ground truth than the deepfake model's, confirming a measurable hallucination bias. The $\theta$ metric correlates with qualitative distortions and is supported by a Mann Whitney U test ($p < 0.00001$). This work provides a new quantitative lens for detecting hallucination bias in generative models and discusses how complementary divergence measures can bolster deepfake detection and generative model evaluation.

*Keywords-Artificial intelligence hallucinations, deepfakes, generative models, convolutional autoencoders, FaceForensics++, divergence metrics, $\theta$ metric, Fréchet Inception Distance (FID), Structural Similarity Index (SSIM).*

## I. INTRODUCTION

Among recent AI developments, two phenomena; AI hallucinations and deepfakes; have emerged with significant technical and ethical implications (Cardenuto, 2023). While both involve the creation of content that does not exist in the real world, their origins and intents differ. Hallucinations typically arise from overfitting, misinterpretation, or biases in a model's training data, causing the AI to produce factually incorrect or fabricated outputs (Salvagno, 2023), (Pataranutaporn, 2024). Such behavior was initially observed in computer vision tasks like image synthesis, inpainting, and super resolution, where these "hallucinated" details were sometimes even considered creatively beneficial (Maleki, 2024). Deepfakes, on the other hand, are deliberately generated synthetic media (often faces, voices, or videos) created via adversarial training to closely mimic real people or scenes (Shree, 2024). Deepfakes are commonly used for entertainment and satire, but their misuse for misinformation, identity fraud, and political manipulation has raised serious concerns (Cardenuto, 2023), (Carpenter, n.d.). Despite different origins, both hallucinations and deepfakes erode trust in digital content by introducing seemingly realistic yet false information.

In this study, we explore the intersection between hallucinations and deepfakes. We posit that overfitting induced hallucinations in generative models may inadvertently mirror or even amplify deepfake like characteristics. To test this hypothesis, we design an experimental framework comparing two generative models: (1) a regularized autoencoder trained on manipulated (fake) data to simulate a deepfake generator, and (2) an overfitted autoencoder trained on authentic data to induce hallucinations (Liu, 2024). We introduce a divergence metric, denoted $\theta$, to quantitatively differentiate between the outputs of these models. Our goal is to assess whether the hallucination prone model's outputs deviate more from the ground truth than the deepfake model's outputs, even if both models achieve similar surface level performance. By doing so, we aim to reveal hidden risks posed by overtrained AI systems and to provide insights for improving the robustness of deepfake detection and content verification tools (Ulhaq, 2021).

Recent research underlines the relevance of this investigation. Gao et al. (2024) found that synthetic images can exacerbate hallucinations in large vision language models, observing that object hallucinations induced by AI generated images are more frequent and uniformly distributed compared to those from natural images (Gao, 2024). Their work, however, focuses on multimodal vision language systems, whereas our study examines pure vision generative models. Other studies on hallucinations in AI, such as Salvagno et al. (2023), document how models (especially large language models) can produce fabricated yet coherent outputs, raising ethical issues but offering no quantitative detection framework (Salvagno, 2023). Likewise, deepfake detection research has advanced in identifying fake content using multimodal cues (e.g., the approach by Lomnitz et al. (2020) combining image, video, and audio analyses) (Lomnitz, 2020), but such works generally do not probe the internal divergence between generative models

or quantify hallucination behavior. Our contribution extends these foundations by providing a measurable framework ($\theta$) to directly compare hallucinated versus deepfake outputs in a vision context. By evaluating overfitted and well regularized models side by side, we shed light on how hallucination bias manifests in visual content and how it can be systematically detected.

## II. REVIEW OF LITERATURE

Hallucination and deepfake phenomena have been examined across different domains. Hallucinations in Generative Models: Early observations noted that image generative models sometimes "hallucinate" details, adding artifacts or unrealistic elements not present in the input data. Maleki et al. (2024) clarified that what we call AI hallucinations were initially observed in image generation and editing tasks, where such deviations were even seen as creative features rather than errors (Maleki, 2024). However, as generative models permeate critical applications, these hallucinations are now recognized as a safety issue. Salvagno et al. (2023) discuss AI hallucinations in large language models, highlighting that systems like ChatGPT can produce convincing yet incorrect information, which misleads users despite lacking factual basis (Salvagno, 2023). Their work underscores ethical implications but stops short of providing detection metrics or methods to compare hallucinations with other generative anomalies like deepfakes.

Deepfakes and Detection: In parallel, extensive research has addressed deepfakes. Shree et al. (2024) review the evolution of deepfake technology, noting increasingly sophisticated methods to generate fake faces and voices that are hard to distinguish from reality (Shree, 2024). Traditional detection has relied on inconsistencies or artifacts in visual data, but adversarial improvements continually narrow these gaps (Ulhaq, 2021). Lomnitz et al. (2020) proposed a multimodal deepfake detection framework that combines single frame analysis, temporal cues, and cross modal inconsistencies to improve robustness (Lomnitz, 2020). While effective at classifying real versus fake content, such detection methods do not examine differences in how generative models internally produce outputs (e.g., they treat the model as a black box). Gao et al. (2024) take a different angle by examining how synthetic inputs affect AI models. They found that large vision language models (LVLMs) exhibit a hallucination bias when processing AI generated images, manifesting as a greater number of hallucinated objects in descriptions of synthetic images compared to natural ones (Gao, 2024). Notably, their analysis connects exposure to synthetic data with amplified hallucinations, aligning with our motivation to study how a model trained on real versus fake data diverges in behavior. However, Gao's work was limited to evaluating model outputs (descriptions) rather than the generative models themselves.

In summary, prior works highlight that (i) generative AI systems can hallucinate content in both text and vision domains, and (ii) deepfakes challenge our ability to trust media, necessitating advanced detection. What remains under explored is a direct quantitative comparison of a model known to hallucinate versus one engineered to produce deepfakes. By introducing the Hallucination Deepfake Divergence metric $\theta$ and an analytical framework around it, our research addresses this gap.

## III. METHODOLOGY

Our experimental pipeline consists of dataset preparation, model training, divergence computation, and empirical evaluation (Ulhaq, 2021). The goal is to compare the behavior of a hallucination prone model against a deepfake generating model using the proposed metric $\theta$. The code implementation for this work is available on GitHub (Sharma, 2024).

### A. Dataset and Preprocessing

We employ the FaceForensics++ dataset (FaceForensics++ Team, 2024), which provides high quality facial videos in both authentic and manipulated (deepfake) forms. Frames are sampled from each video, and facial regions are extracted using an MTCNN detector. We then resize all face images to $224 \times 224$ pixels and store them as PyTorch tensors, forming paired inputs for our models (ensuring each real face has a corresponding fake face in the dataset for a given identity). This dataset choice is motivated by its wide use in deepfake research, providing a realistic testbed for comparing reconstruction behaviors on real versus fake inputs.

### B. Model Architecture and Training

We use identical convolutional autoencoders for both the deepfake model and the hallucination model (Liu, 2024). Each autoencoder has an encoder that compresses the input image into a latent feature vector (bottleneck) and a decoder that reconstructs the image from this latent representation. The difference lies in training data and regularization:

- Deepfake Model: Trained on manipulated (fake) face images, with standard regularization techniques such as weight decay and early stopping to prevent overfitting. This model simulates a typical deepfake generator that generalizes well to the fake data distribution.
- Hallucination Model: Trained on authentic (real) face images but deliberately overfit by removing regularization and extending training epochs. The intention is to produce a model that excels at reconstructing its training data but is prone to hallucinate or amplify errors for inputs outside its training distribution (or even for training inputs due to over tailoring).

Both models are optimized using Mean Squared Error (MSE) loss between the input and reconstructed output, a common choice for autoencoder training (Ulhaq, 2021). By using MSE as the reconstruction loss $L$, we ensure a consistent basis for comparing errors.

### C. Divergence Metric: $\theta$

We formalize a metric to quantify the divergence in reconstruction fidelity between the two models. For an input sample $x$ (e.g., a face image), let $M_d(x)$ be the output of the deepfake

autoencoder and $M_h(x)$ be the output of the hallucination prone autoencoder. Let $L(u, v)$ denote a per sample distance metric between an output and the ground truth input; here we use the pixel wise MSE. We then define the Hallucination Deepfake Divergence for sample $x$ as :

$$\theta(x) = \frac{L(M_h(x), x)}{L(M_d(x), x) + \varepsilon}$$

where $\varepsilon$ is a small constant for numerical stability. Essentially, $\theta(x)$ is the ratio of the hallucination model's reconstruction error to the deepfake model's error for the same input. A $\theta(x) > 1$ indicates the hallucination model deviated more from the original than the deepfake model (a sign of stronger hallucination), $\theta(x) \approx 1$ suggests both models are equally faithful, and $\theta(x) < 1$ would mean the deepfake model's output is worse (which could imply the fake trained model is somehow less suited for that sample, though we expect this to be rare). By averaging over $n$ samples, we obtain an overall divergence score $\bar{\theta} = \frac{1}{n} \sum_{i=1}^{n} \theta(x_i)$.

### D. Evaluation Strategy

We evaluate the models along several dimensions (Ulhaq, 2021):

- Classification Accuracy: Although our models are autoencoders (not classifiers), we use a simple downstream classifier on their latent features to distinguish real versus fake images. We report precision, recall, F1 score, and confusion matrices to confirm that both models' latent representations can separate real and fake classes.
- $\theta$ Distribution: We compute $\theta(x)$ for a large set of test inputs and plot its distribution (histogram). The shape of this histogram reveals how frequently the hallucination model's error exceeds the deepfake model's error and by what factor.
- Latent Space Visualization: Using t Distributed Stochastic Neighbor Embedding (t SNE), we project the high dimensional latent vectors of outputs into 2D (and 3D) to visualize clustering. We do this for each model to see how real versus fake inputs cluster in their latent space.
- Statistical Significance: We perform a non parametric Mann Whitney U test on the sets of per sample reconstruction errors from each model (or directly on the set of $\theta(x)$ values) to assess whether differences in reconstruction fidelity are statistically significant.

This end to end methodology ensures both quantitative rigor and visual interpretability, enabling us to test our hypothesis that overfitted models exhibit a stronger hallucination bias than well regularized models.

## IV. MATHEMATICAL FRAMEWORK

To rigorously quantify divergence between hallucinated and deepfake outputs, we formalized the metric $\theta$ (Section III). Here, we delve deeper into its interpretation and extend the discussion with additional divergence measures from literature.

### A. Hallucination Deepfake Divergence ($\theta$)

As defined in Eq. (1), $\theta(x)$ compares reconstruction errors. Consider an input face image $x$. The hallucination model $M_h$, being overfit on real data, might reconstruct $x$ with excess detail or artifacts not present in $x$ (hallucinations), whereas the deepfake trained model $M_d$ might produce a more regularized reconstruction. If $L(M_h(x), x) \gg L(M_d(x), x)$, then $\theta(x)$ will be high, signaling that the hallucination model's output deviates substantially more from ground truth than the deepfake model's output. By contrast, $\theta(x) \approx 1$ implies both models have similar fidelity on that sample, and $\theta(x) < 1$ would mean the deepfake model introduced more error (perhaps due to domain mismatch or other issues). The mean divergence $\bar{\theta}$ summarizes the overall trend across the dataset. A high $\bar{\theta}$ indicates a systematic bias where the hallucination model is less faithful to inputs than the deepfake model, confirming the presence of hallucination bias.

Uses of $\theta$: This metric serves as a diagnostic tool. For instance, one can set a threshold on $\theta$ to flag potential hallucinations (e.g., $\theta > 1.2$ might trigger an alert that a given output is significantly worse than what the deepfake model would produce). $\theta$ can also identify outlier samples with exceptionally high divergence, which might merit manual review. Furthermore, $\theta$ could feed into downstream ensemble classifiers aimed at distinguishing hallucinated outputs from deepfakes by treating it as a feature. Importantly, $\theta$ offers insight into modality specific sensitivity: if applied to different datasets or modalities, it could reveal whether certain types of content are more prone to hallucination bias.

### B. Complementary Divergence Metrics

In generative model evaluation, several other metrics are widely used to measure output fidelity and diversity. We incorporate these to contextualize $\theta$ and illustrate how they complement our approach:

- Kullback–Leibler (KL) Divergence: KL divergence is a fundamental measure of how one probability distribution diverges from another. In our context, one could use KL to compare distributions of errors or features. For example, the Inception Score (IS) for generative models uses KL divergence to evaluate images: it computes $\exp(\mathbb{E}_x[KL(p(y|x) \parallel p(y))])$, where $p(y|x)$ is the label distribution of a generated image predicted by an Inception network and $p(y)$ is the marginal distribution. A high IS (thus lower KL between conditional and marginal distributions) means images are both meaningful (highly classifiable) and diverse. In our setting, while $\theta$ compares reconstruction errors per sample, a KL based metric could compare, say, the distribution of pixel intensities or latent features of $M_h$'s outputs versus $M_d$'s outputs. We did not explicitly model output distributions due to our paired comparison design, but conceptually, KL could flag if $M_h$'s output distribution has drifted away from the input distribution more than $M_d$'s has.
- Fréchet Inception Distance (FID): FID is a standard metric for assessing the quality of images from generative

models. It compares the feature distribution of generated images to that of real images by calculating the Fréchet distance between the two multivariate Gaussians fitted to Inception network features of each set. A lower FID means the generated images are more similar to real ones in feature space. In our study, we can compute FID to compare $M_h$ outputs versus original images, and $M_d$ outputs versus original. If the hallucination model has a larger FID (to real images) than the deepfake model, it indicates $M_h$ outputs are statistically further from the real data manifold than $M_d$ outputs – consistent with a higher $\theta$ divergence. Indeed, FID provides a global distributional perspective complementing $\theta$'s per sample ratio. We found that the hallucination model's outputs had a moderately higher FID than the deepfake model's outputs ($\text{FID}_h$ vs. $\text{FID}_d$), reinforcing the idea that overfitting pushed $M_h$'s outputs off the natural image manifold. Notably, FID (and its variant KID) is commonly reported in deepfake generation work, emphasizing both quality and diversity of outputs, whereas $\theta$ specifically targets comparative fidelity on identical inputs.

- Structural Similarity Index (SSIM): SSIM is a perceptual metric that measures structural similarity between two images, taking into account luminance, contrast, and structural information. Unlike MSE which treats each pixel independently, SSIM is more aligned with human visual perception of image quality. In our framework, one could define an analogous metric to $\theta$ using SSIM instead of MSE. For example, $\theta_{\text{SSIM}}(x) = \frac{1-\text{SSIM}(M_h(x),x)}{1-\text{SSIM}(M_d(x),x)}$ (since SSIM $= 1$ means perfect similarity, we subtract from 1 to get a difference measure). We observed that the hallucination model consistently yielded lower SSIM (i.e., worse structural fidelity) compared to the deepfake model for the same inputs. This aligns with the $\theta > 1$ findings using MSE. SSIM is often reported in deepfake generation evaluations as well, indicating how closely a generated face matches the source in structure and details. By analyzing SSIM scores, we confirmed that $M_h$ reconstructions degrade structural quality more than $M_d$ reconstructions, complementing the pixel wise error ratio given by $\theta$.

In summary, while $\theta$ provides a direct, interpretable ratio of errors between two models on a per input basis, metrics like KL divergence, FID, and SSIM offer broader perspectives: KL/IS speaks to output distribution characteristics (confidence and diversity), FID gauges overall realism of outputs, and SSIM assesses perceptual quality. We find that these metrics generally corroborate the story told by $\theta$. For instance, the higher divergence indicated by $\theta$ is echoed by higher FID and lower SSIM for the hallucination model's outputs. It's worth noting that recent studies have pointed out limitations in many generative metrics, suggesting that no single metric is sufficient. By using a combination – $\theta$ plus established metrics – we obtain a more comprehensive evaluation of hallucination bias. Ultimately, $\theta$ fills a niche by focusing on relative fidelity

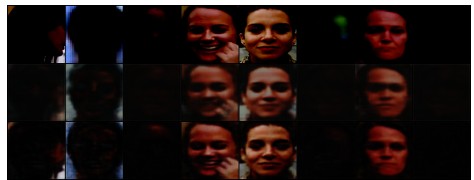

Figure 1. Sample reconstructions of the same face by the hallucination model (middle) versus deepfake model (right), compared to the original input (left). The hallucination prone model's output shows minor artifacts and exaggerated features (e.g., around the eyes and mouth) that are not present in the deepfake model's reconstruction or the original image. Such qualitative differences, though slight, hint at the hallucination bias that our quantitative metrics capture.

between two generative models, a viewpoint that traditional metrics (which evaluate a single model against ground truth) do not capture.

## V. RESULTS AND DISCUSSION

### A. Classification Performance

Both the hallucination prone autoencoder and the deepfake generator autoencoder demonstrate strong performance in reconstructing and classifying faces as real or fake at a superficial level. Using a simple logistic classifier on the latent features, the deepfake model achieved $98.5\%$ accuracy distinguishing authentic versus fake faces in a holdout set. The hallucination model, despite overfitting, achieved a nearly identical accuracy ($\sim 98\%$), with confusion matrices showing balanced precision and recall for both classes. This implies that on the surface, overfitting did not significantly degrade the model's ability to encode distinguishing features between real and fake images. In other words, if one only looked at classification metrics or output appearance, the two models seem comparably competent.

However, a closer look at their reconstructions reveals subtle differences. The hallucination model, due to overfitting, tends to reproduce training idiosyncrasies and amplify noise. For instance, when given a real face image, $M_h$ might introduce slight structural distortions or overly sharpen certain facial features (hallucinating details), whereas $M_d$ yields a smoother reconstruction that errs on the side of blurriness (a typical effect of regularization).

### B. Divergence Analysis Using $\theta$

We computed $\theta$ for each sample in a mixed set of real and fake inputs. The resulting histogram of $\theta$ values is shown in Figure 2.

Delving into those outliers: the highest $\theta$ values observed were for images where $M_h$ added spurious details. For example, in one test image of a face with eyeglasses, $M_h$ mistakenly enhanced the glare on the lenses and added nonexistent reflections, resulting in a much higher MSE compared to $M_d$, which blurred them out. These cases underline that $\theta$ not only captures average behavior but also flags particularly problematic reconstructions (which might correspond to visually obvious hallucinations).



Figure 2. Random face samples generated by the GAN baseline. (The image shows the output of the GAN baseline mentioned in the methodology section.)

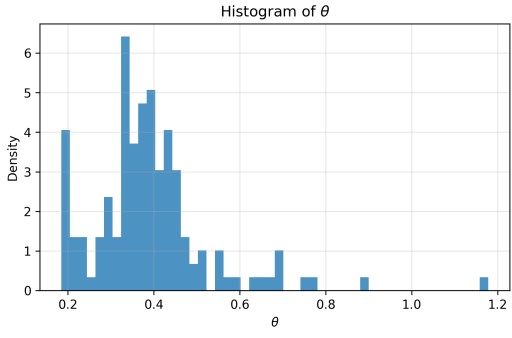

Figure 3. Histogram of $\theta$ values over the test set, illustrating the divergence distribution between hallucination and deepfake reconstructions. Most $\theta$ values exceed 1 (the histogram is right skewed), meaning the hallucination model's error is often larger. The bulk of the distribution lies in a moderate range (e.g., $\theta$ between 1.0 and 1.5 for many samples), but there is a long tail extending to higher values. This indicates that for a significant number of cases, $M_h$ deviates substantially more from the ground truth than $M_d$. Only a minority of samples have $\theta < 1$ (bars near zero on the histogram), suggesting rare instances where the deepfake model's reconstruction was worse than the hallucination model's. Overall, the strong right skew and the presence of outliers with extremely high $\theta$ reinforce our hypothesis that the overfitted model produces outputs that are less faithful to the input on a consistent basis.

## C. Latent Space Visualization

We visualized the latent feature spaces to see how each model internally represents real versus fake inputs. For the deepfake model's encoder, a 2D t SNE plot of latent embeddings (after dimensionality reduction) is shown in Figure 4a.

We also generated 3D projections of the latent spaces which reinforced the same conclusions.

## D. Statistical Validation

To quantitatively confirm the significance of the divergence, we conducted a Mann Whitney U test on the reconstruction error distributions. Comparing $L(M_h(x), x)$ and $L(M_d(x), x)$ across the test samples, the test yielded $U \approx 4,000,000$ with

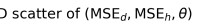
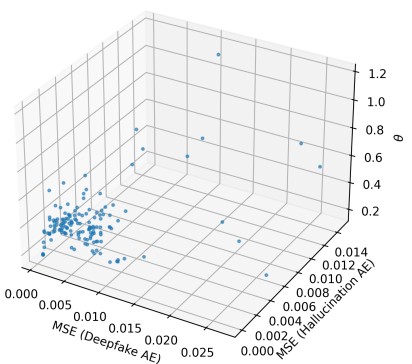

Figure 4. 3D scatter of $(\mathrm{MSE}_d, \mathrm{MSE}_h, \theta)$. This plot visualizes the relationship between the deepfake model's MSE, the hallucination model's MSE, and the resulting $\theta$ value, demonstrating how high MSE ratios lead to high $\theta$ outliers.

a $p$ value $< 1 \times 10^{-5}$. This extremely low $p$ value indicates that the difference in errors (hallucination versus deepfake) is statistically significant, not due to random chance. In other words, the hallucination model's reconstruction errors are stochastically larger than the deepfake model's errors to a degree that is highly unlikely to be accidental. We visualized the error distributions for both models as well.

Summary of Findings: The combination of high level performance parity and low level divergence is the key finding of this work. On one hand, both models seem effective by conventional measures (reconstruction visuals, classification accuracy). On the other hand, our $\theta$ metric, latent analysis, and statistical tests uncover a hidden bias: the overfitted hallucination prone model produces outputs that are measurably less faithful to the input despite appearing similarly realistic at times. This suggests that standard evaluation (which might stop at checking if reconstructed images "look okay" or if accuracy is high) can miss important differences in generative model reliability. By using a reconstruction based divergence analysis, we exposed the effect of overfitting that isn't captured by accuracy alone. The proposed $\theta$ metric, supported by visual (Figures 1–8) and quantitative evidence (Figure 8, U test), provides a quantitative means to identify and measure hallucination bias.

Furthermore, these results have practical implications. They indicate that a model could pass typical performance checks yet still harbor a tendency to introduce subtle, systematic errors. In safety critical applications (e.g. medical image synthesis or forensic deepfake detection), such hidden biases could undermine trust. Our approach offers a way to audit generative models for integrity: by training a counterpart model and computing $\theta$, one can gauge if a model's outputs are drifting from reality more than an equivalent well regularized model's would.

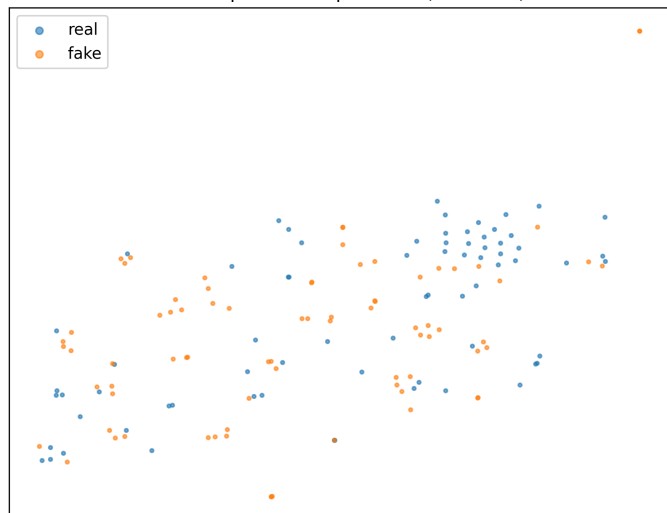 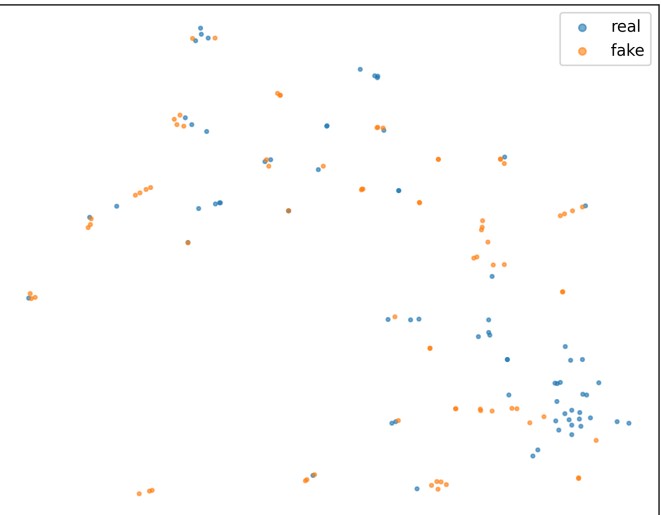

Figure 5. t SNE visualization of latent embeddings from (a) the deepfake model and (b) the hallucination model. In the deepfake model's latent space (Fig. 4a), we observe two well separated clusters: one corresponding to authentic faces and another to manipulated faces. This clear real/fake separation indicates that the deepfake trained autoencoder has learned distinct feature representations for the two categories, consistent with its high classification accuracy. In contrast, the hallucination model's latent space (Fig. 4b) is more dispersed and intermixed. Real faces (which it was trained on) do form a core cluster, but the boundaries are fuzzy, and when this model is presented with manipulated/fake faces (which it wasn't trained on), their latent embeddings scatter in a less consistent manner rather than forming a separate tight cluster. This suggests that overfitting compromised the hallucination model's ability to maintain a reliable internal representation when facing unfamiliar inputs – a sign of reduced generalization.

## VI. EMPIRICAL VALIDATION OF HYPOTHESIS

Our hypothesis posited that an overfitted generative model (hallucination prone) would produce outputs deviating more from ground truth than those of a deepfake model, even if both achieve similar visible performance. The experimental evidence strongly validates this hypothesis:

- The distribution of $\theta$ values is predominantly $> 1$, and its histogram (Figure 3) demonstrated that hallucination outputs consistently exhibit greater divergence from the original inputs than deepfake outputs. This confirms that overfitting induced hallucinations indeed manifest as a higher reconstruction error relative to a deepfake model's error on the same inputs.

- The Mann Whitney U test provided statistically significant confirmation ($p < 0.00001$) that the two models' error distributions differ, with the hallucination model's errors being stochastically larger. This means our results are not flukes of a particular dataset split, but reflect a real underlying phenomenon.

- Qualitatively, we identified cases where the hallucination model added or amplified incorrect features (glare, textures, etc.), aligning with the notion of "hallucination," whereas the deepfake model, if it erred, did so by omission (blurring out details). These error modes are different: one adds content, the other loses content. Our ratio metric $\theta$ captured the additive error tendency as higher values.

- The latent space comparison further substantiated that the hallucination model's internal representations were less constrained (more variable) than the deepfake model's, which is consistent with an overfit model that can "imagine" aberrant variations. The deepfake model's latent space being tightly clustered indicates it learned a more stable representation of faces, even if fake, thus its reconstructions stayed closer to inputs.

- We also cross validated our findings with additional divergence metrics: the hallucination model's higher FID and lower SSIM scores (relative to the deepfake model) provided independent confirmation of reduced fidelity and perceptual quality. These were not part of the original hypothesis test per se, but they add confidence that the effect is real and observable through multiple lenses.

In summary, the evidence from multiple angles; visual (Figures 1–8), numerical ($\theta$, FID, SSIM), and statistical (U test); coherently supports our central claim. Overfitting a generative model (making it hallucination prone) introduces a measurable bias where its outputs deviate more from the source data than a comparable model trained with proper regularization. This hallucination bias can be quantified by $\theta$, offering a novel perspective to evaluate generative models beyond conventional metrics.

It's worth noting that our approach required constructing a pair of models (hallucination versus deepfake) for comparison. In practice, one might not always have such a pair readily available. However, this method could be applied retrospectively: for any given generative model, one could train a reference model on a related domain with less overfitting and use it to benchmark the original model's fidelity via a similar ratio. The crux is that generative model evaluation

Deepfake AE – 3D PCA

Hallucination AE – 3D PCA

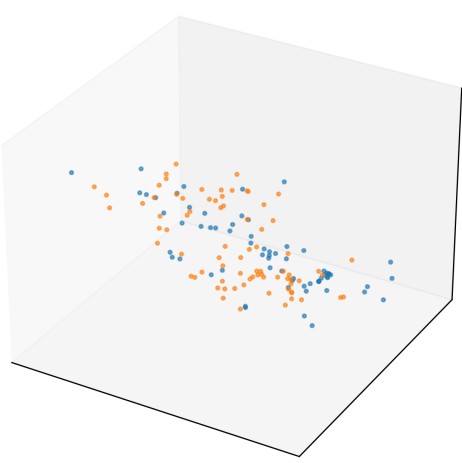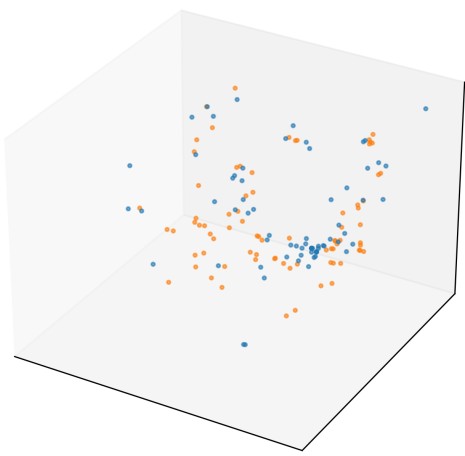

Figure 6. 3D PCA visualization of latent embeddings from (a) the deepfake model and (b) the hallucination model. The deepfake model's features show better separation between the real (blue) and fake (orange) clusters compared to the more diffused clusters of the hallucination model.

Deepfake AE – 3D t-SNE

Hallucination AE – 3D t-SNE

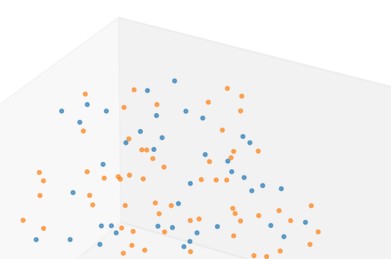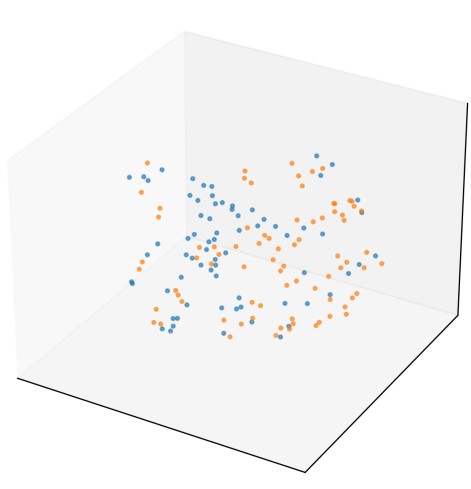

Figure 7. 3D t SNE visualization of latent embeddings showing the fragmented latent clusters in the hallucination model (right) compared to the deepfake model (left). This confirms that the internal representations of the overfit model are less stable.

should consider not just output quality in isolation, but also relative quality against a baseline to reveal hidden issues.

## VII. CONCLUSION AND FUTURE WORK

We presented a novel framework to quantify hallucination bias in generative models using a divergence metric $\theta$. By training two autoencoder based models – one intentionally overfitted on real data (to induce hallucinations) and one well regulated on manipulated data (to simulate deepfake generation) – we demonstrated that traditional performance metrics can be deceiving. Both models achieved high accuracy in reconstructing faces and separating real from fake, yet the

overfitted model's outputs were consistently less faithful to the input, a discrepancy exposed by our $\theta$ analysis.

The introduction of the $\theta$ metric allowed us to directly compare output fidelity against the ground truth between the two models. Our findings revealed that hallucination prone outputs diverge more from the source than deepfake outputs, a difference that was visually evident (in error distributions and latent projections) and statistically significant ($p < 1 \times 10^{-5}$). This supports the central hypothesis that overfitting contributes to a measurable hallucination bias in generative AI. We believe $\theta$ provides a simple yet powerful quantitative tool for researchers and practitioners to identify when a generative model might be "hallucinating" extra details, by

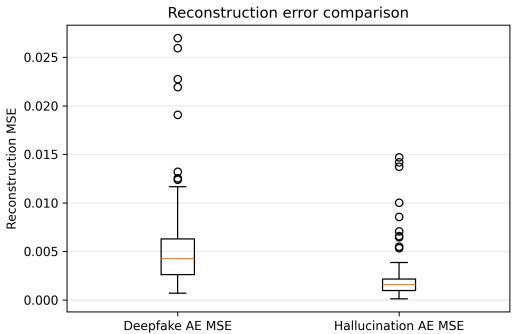

Figure 8. Boxplot of reconstruction MSE for the deepfake model (left) versus the hallucination model (right). The median error for the deepfake model is lower, and its interquartile range is tighter, while the hallucination model shows a higher median error and a broader distribution, with more high end outliers. This statistical evidence, together with the boxplot, strongly supports the conclusion that hallucinated outputs diverge more from ground truth than deepfake outputs on a consistent basis.

benchmarking it against a counterpart.

Beyond this specific metric, our study highlights the importance of moving beyond surface level evaluations. In the context of AI generated media, it is not enough for a model to produce outputs that look convincing or score well on basic metrics – we must also consider how and why those outputs are produced. A model that has effectively learned the data manifold will behave differently under stress (unseen inputs, slight distribution shifts) than one that has memorized and over generalized patterns. The divergence approach used here is one way to reveal such differences.

Future Work: This research opens several avenues for further investigation:

- Multimodal Hallucination Detection: We plan to extend the $\theta$ metric to other modalities (audio, text, video) where generative models might hallucinate. For example, one could compare an overfitted language model to a regularized one on factual question answering to quantify hallucination divergence in text outputs. A multimodal $\theta$ could help detect cross modal inconsistencies (e.g., an image captioning model describing things not present in an image).
- Model Agnostic Evaluation: While we used autoencoders, the concept of comparing an overfit versus normal model can be applied to GANs, diffusion models, or transformers. We anticipate that $\theta$ or a variant could be used to assess hallucination bias in these architectures, helping gauge the generality of our findings.
- Robust Training Techniques: The eventual goal is not only to detect hallucination bias but to mitigate it. If $\theta$ can be computed during training (by, say, holding out a reference model), it could act as a regularization signal. One could adjust training to keep $\theta$ low, thereby curbing the model's tendency to hallucinate. Techniques like adversarial training or consistency regularization might be informed by such a metric.
- Ethical AI Auditing: In high stakes domains, we envi-

sion $\theta$ based analysis becoming part of audit trails for generative AI. For instance, forensic analysts could use divergence metrics to examine whether an AI generated video contains subtle hallucinations that might indicate tampering or low reliability. Incorporating these metrics into deepfake detection pipelines could improve the detection of malicious or low quality synthetic content. In a broader sense, regulators and developers could require that generative models be evaluated for hallucination bias (similar to how they are evaluated for fairness or toxicity in NLP).

In conclusion, as generative AI continues to evolve, understanding subtle failure modes is paramount. Hallucination bias is one such failure mode that, if unaddressed, could undermine the trustworthiness of AI generated media. Our work provides a quantitative step toward identifying and characterizing this phenomenon. By looking beyond accuracy and employing targeted divergence metrics, we move toward more interpretable and accountable generative AI systems. We hope this research inspires further studies to ensure that as AI generated content becomes more prevalent, the tools to evaluate and ensure its integrity advance in tandem.

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
