# OpenReview forum: "Quantifying Hallucination Bias in AI-Generated Deepfakes: A Multimodal Analysis Using Divergence Metrics"
_PAKDD.org/2026/Workshop/JEN-AI — PAKDD 2026 Workshop JENAI Withdrawn Submission_

### Official Review · Reviewer_hJgp · 2026-03-10
**The θ metric is an MSE ratio — even with relaxed workshop expectations, the confound and missing numbers are hard to overlook**

**Rating:** 4
**Confidence:** 4

**Review:**

The paper proposes comparing two convolutional autoencoders — one deliberately overfit on real faces (the "hallucination model") and one trained with regularization on manipulated faces (the "deepfake model") — and introduces $\theta$, defined as the ratio of their reconstruction MSEs for the same input. The claim is that this provides a novel framework for detecting hallucination bias in generative models, evaluated on FaceForensics++.

I've recalibrated my expectations for this venue — JEN-AI is a first-edition workshop co-located with PAKDD, accepting 4–6 page short papers and encouraging exploratory work at the intersection of generative AI and journalism. Even with that more generous lens, I have significant concerns about this submission.

## What I found interesting

The observation that both models achieve ~98% classification accuracy while diverging substantially in reconstruction fidelity is a useful pedagogical point: surface-level metrics can mask important differences in generative model behavior. The qualitative examples — particularly the eyeglasses case where the overfit model hallucinated lens reflections — illustrate this well. The general idea of benchmarking a model against a reference counterpart to reveal hidden fidelity issues has some appeal as an auditing concept.

## Where this falls short, even for a workshop paper

**The $\theta$ metric doesn't offer enough novelty for any venue.** It's $\theta(x) = L(M_h(x), x) \,/\, (L(M_d(x), x) + \varepsilon)$. Dividing one model's error by another's is a standard sanity check, not a research contribution. The paper devotes a full "Mathematical Framework" section to this, but there's no information-theoretic grounding, no connection to latent geometry, no probabilistic interpretation. The discussion of KL, FID, and SSIM reads like a textbook survey rather than an integration — and crucially, the actual FID and SSIM numbers are never reported, just described qualitatively ("moderately higher," "consistently lower"). For a quantitative paper, this is a significant gap.

**The experimental design has a fatal confound.** The two models differ along *two* axes simultaneously: training data (real vs. manipulated) and regularization (overfit vs. regularized). When the overfit model shows higher error, is that "hallucination bias" or just "we made this model worse on purpose"? Even for exploratory workshop work, you need to isolate the variable you're studying. Train both on the same data with different regularization, or both with the same regularization on different data. The current design can't support the causal claim.

**The terminology is stretched.** Calling higher reconstruction MSE from an overfit autoencoder "hallucination bias" conflates reconstruction degradation with the ML community's understanding of hallucination (generating confident, plausible-but-false content). The $\theta$ metric can't distinguish between additive hallucinations and generic noise amplification. This isn't a nitpick — the whole paper hinges on this connection being meaningful.

**The title claims "multimodal" but only images are used.** The authors mean "multiple metrics," not "multiple modalities." That's a misuse of the term and should be corrected regardless of venue.

## Fit for this workshop

I also want to flag a scope concern. JEN-AI is about generative AI and computational journalism. This paper is about autoencoder reconstruction quality on face images. The connection to journalism is thin — the paper gestures toward deepfake detection for forensic purposes, but the actual experiments are purely a computer vision exercise comparing two autoencoders. I'm not sure this fits the workshop's scope.

## Overall assessment

Even with relaxed workshop expectations, the core technical contribution (an MSE ratio) is too thin, the experimental confound is unaddressed, the quantitative details are missing, and the fit to this workshop's theme is unclear. I'd encourage the authors to fix the confound, report actual numbers for all metrics, clarify the hallucination terminology, and consider targeting a computer vision workshop where the contribution would be more naturally situated.

**Confidential Comments To Pc:**

Metric is trivially an MSE ratio, experiment confounds training data with regularization, FID/SSIM numbers never reported, "multimodal" label is wrong, and fit to journalism theme is unclear. Raised from 3 to 4 for workshop expectations, but the confound alone is disqualifying—central claim doesn't follow from the experiment.

**Ethical Concerns:**

1

---

### Official Review · Reviewer_HWRZ · 2026-03-11
**Interesting premise, but the proposed metric is shallow and the experimental design does not support the main claims**

**Rating:** 3
**Confidence:** 4

**Review:**

This paper proposes a metric called theta to compare reconstruction errors between an overfit autoencoder trained on real faces and a regularized autoencoder trained on manipulated faces from FaceForensics++. The stated goal is to quantify whether overfitting induced hallucinations produce outputs that deviate from ground truth more than deepfake model outputs. While the research question is reasonable, the paper has several fundamental issues that prevent it from making a meaningful contribution.
Major Issues

The proposed theta metric is simply the ratio of two MSE values. Defining theta = MSE_h / MSE_d tells us that one model has higher reconstruction error than the other. This does not yet rise to the level of a substantive methodological contribution. It is a ratio. The paper spends considerable space (Sections III.C and IV.A) formalizing and interpreting this quantity, but there is nothing here that goes beyond what you would get from just reporting the two MSE distributions side by side and running a standard comparison. Framing a ratio of losses as a novel contribution overstates the actual technical novelty.
The experimental design conflates two variables and draws conclusions that do not follow. The hallucination model is trained on real data and deliberately overfit. The deepfake model is trained on fake data with regularization. So we have two differences at once: the training data domain and the regularization regime. When theta exceeds 1, the paper interprets that as evidence of hallucination bias, but that inference is not supported by the current design. But this could just as easily reflect domain mismatch, since the overfit model trained on real faces is being asked to reconstruct both real and fake inputs, while the regularized model trained on fakes may simply be better calibrated to the test distribution. Without ablations that isolate these factors, for example an overfit model on fake data, or a regularized model on real data, the causal claim about overfitting causing hallucination bias is not supported. This is a basic experimental design issue.
The paper claims that both models achieve similar classification accuracy (98% vs 98.5%) and then presents divergence in theta as a hidden problem that surface metrics miss. But the classification task here is performed by a simple logistic classifier on the latent features, not by the autoencoder itself, and the accuracy numbers are suspiciously high for such a simple setup. The paper does not report what test set was used for classification, how many samples, or whether there is any overlap with training data. Without these details, the high accuracy numbers are hard to interpret, and the framing that standard metrics are deceiving rings hollow.
The connection between what the paper calls hallucination and what the broader community means by the term is quite loose. In the generative AI literature, hallucination typically refers to models producing semantically coherent but factually wrong content, for example fabricated entities in text or plausible but nonexistent objects in images. What this paper calls hallucination is really just reconstruction artifacts from an overfit autoencoder. These are standard overfitting artifacts that have been understood for decades. Calling them hallucinations and then claiming a link to the deepfake problem requires much stronger justification than what is provided. Gao et al. 2024 (cited as [10]) study a genuinely different phenomenon, how synthetic images induce object level hallucinations in vision language models, and the connection to the present work is superficial.
The paper only uses convolutional autoencoders. Modern deepfake generation relies on GANs (StyleGAN, ProGAN), diffusion models, and encoder decoder architectures far more sophisticated than a vanilla convolutional autoencoder. The FaceForensics++ dataset itself contains forgeries generated by DeepFakes, Face2Face, FaceSwap, and NeuralTextures as documented in Rossler et al. at ICCV 2019. Using a simple autoencoder trained on the manipulated subset and calling it a deepfake model is a significant oversimplification. The results may not generalize to any realistic deepfake generation scenario.
The statistical testing (Mann Whitney U, p < 0.00001) confirms that two different models trained on different data with different regularization produce different error distributions. This is entirely expected and does not validate the theta metric or the hallucination bias hypothesis. The null hypothesis being rejected here is so weak that the test adds almost no information.

Minor Issues

The FaceForensics++ dataset citation is listed as "FaceForensics++ Team (2024). Kaggle Dataset." The original and proper citation is Rossler et al. ICCV 2019. Citing a Kaggle mirror rather than the original publication is not appropriate.
Reference [8] (Ulhaq, 2021) is an engrXiv preprint that provides a general deep learning survey. It is cited three times in the methodology section to justify standard choices like using MSE loss and standard evaluation strategies. These are textbook decisions that do not need citation support from a preprint, and the repeated citation of this particular paper for routine methodology choices is unusual.
The complementary metrics section (IV.B) discusses KL divergence, FID, and SSIM at length but never actually reports their numerical values. The paper says it "found" that FID was higher for the hallucination model, but no numbers are given. Similarly for SSIM. This makes the discussion read more like a literature review of metrics than an actual empirical evaluation.
Figure 2 shows "random face samples generated by the GAN baseline" according to its caption, but the methodology section does not describe training any GAN. This figure appears disconnected from the rest of the paper.
Figure 3 appears visually inconsistent with the accompanying textual claim that most theta values exceed 1. The authors should clarify the binning, scaling, and summary statistics.


Summary
The idea of connecting overfitting artifacts in generative models to deepfake detection is a potentially interesting direction. However, the proposed theta metric lacks technical depth, the experimental design confounds training domain with regularization, key metrics are discussed but not reported, and there is a visible inconsistency between the histogram and the claims made about it. The paper would benefit substantially from proper ablations, realistic generative architectures, and actual numerical reporting of all claimed metrics.

**Confidential Comments To Pc:**

Figure 3 appears to contradict the text claims about theta distribution. The histogram shows most mass below 1.0, yet the paper states most values exceed 1. This warrants attention. Reference practices are also unusual, with a Kaggle mirror cited instead of Rossler et al. ICCV 2019, and a single engrXiv preprint cited three times for routine methodology. Beyond presentation issues, my main reason for rejection is that the core experiment does not isolate the effect it claims to study.

**Ethical Concerns:**

1

---

### Note · Authors · 2026-03-11

I have read and agree with the venue's withdrawal policy on behalf of myself and my co-authors.